# The Preparation and Characterization of Quinoa Protein Gels and Application in Eggless Bread

**DOI:** 10.3390/foods13081271

**Published:** 2024-04-21

**Authors:** Qianqian Xu, Xinxia Zhang, Zhongyu Zuo, Ming Zhang, Ting Li, Li Wang

**Affiliations:** 1National Engineering Research Center for Cereal Fermentation and Food Biomanufacturing, Jiangnan University, Lihu Road 1800, Wuxi 214122, China; xuqianqian1119@126.com (Q.X.); 8201906115@jiangnan.edu.cn (X.Z.); zuozhongyu2016@outlook.com (Z.Z.); zm911927856@163.com (M.Z.); ting.li@jiangnan.edu.cn (T.L.); 2Jiangsu Provincial Engineering Research Center for Bioactive Product Processing, Jiangnan University, Lihu Road 1800, Wuxi 214122, China; 3School of Food Science and Technology, Jiangnan University, Lihu Road 1800, Wuxi 214122, China; 4Key Laboratory of Carbohydrate Chemistry and Biotechnology, Ministry of Education, Jiangnan University, Lihu Road 1800, Wuxi 214122, China; 5State Key Laboratory of Food Science and Resources, Jiangnan University, Lihu Road 1800, Wuxi 214122, China

**Keywords:** quinoa protein gels, ultrasound, eggless bread

## Abstract

The properties of xanthan gum protein gels composed of quinoa protein (XG-QPG) and ultrasound-treated quinoa protein (XG-UQPG) were compared for the preparation of high-quality quinoa protein gels. The gel qualities at different pH values were compared. The gels were used to produce eggless bread. Microscopically, the secondary structure of the proteins in XG-QPG (pH 7.0) was mainly α-helix, followed by random coiling. In contrast, the content of β-sheet in XG-UQPG was higher, relative to the viscoelastic properties of the gel. Moreover, the free sulfhydryl groups and disulfide bonds of XG-QPG (pH 7.0) were 48.30 and 38.17 µmol/g, while XG-UQPG (pH 7.0) was 31.95 and 61.58 µmol/g, respectively. A high disulfide bond content was related to the formation of gel networks. From a macroscopic perspective, XG-QPG (pH 7.0) exhibited different pore sizes, XG-UQPG (pH 7.0) displayed a loose structure with uniform pores, and XG-UQPG (pH 4.5) exhibited a dense structure with small pores. These findings suggest that ultrasound can promote the formation of a gel by XG-UQPG (pH 7.0) that has a loose structure and high water-holding capacity and that XG-UQPG (pH 4.5) forms a gel with a dense structure and pronounced hardness. Furthermore, the addition of the disulfide bond-rich XG-UQPG (pH 7.0) to bread promoted the formation of gel networks, resulting in elastic, soft bread. In contrast, XG-UQPG (pH 4.5) resulted in firm bread. These findings broaden the applications of quinoa in food and provide a good egg substitute for quinoa protein gels.

## 1. Introduction

Bread is an important component of many diets and is a staples of the Western diet [1]. However, a diet based on wheat bread may not be the most effective approach as it lacks high-quality protein and lysine [2]. Adding other sources of protein to the product formulation is an approach used to increase the nutritional value of wheat products, such as adding eggs to supplement egg white protein. While eggs are rich in cholesterol and common food allergens, their consumption does not align with the principles of green and low-carbon living [3]. This gas retention is crucial for bread-making, significantly enhancing texture and volume. Additionally, the use of eggs in gluten-free bread formulations has been shown to substantially elevate product quality [4,5]. Despite these benefits, the environmental and health concerns associated with egg consumption—namely high cholesterol content and allergenic potential—prompt the exploration of sustainable alternatives [6]. 

In response to these concerns, researchers have investigated alternative ingredients that mimic the functional properties of eggs in bread. One promising substitute is the protein–polysaccharide complex gel, known for its ability to improve dough properties and achieve physicochemical characteristics akin to those of traditional bread. Hydrocolloids, such as alginate (ALG), xanthan gum (XG), carrageenan, and *Morinda officinalis* polysaccharides (MOP), play a pivotal role in this context by enhancing dough development and gas retention through increased viscosity, resulting in bread with superior baking qualities [7,8,9]. Furthermore, compared to the addition of protein alone, incorporating soy protein–polysaccharide hydrophilic colloids produces a batter with similar specific gravity and viscosity similar to that of egg cake batter [10]. However, a comprehensive exploration and their structural characterization have not been extensively investigated, indicating a vast area for future research.

With the growing consumer demand for nutritious and healthy foods, there is an increasing trend of substituting animal protein with plant proteins. Plant proteins provide comprehensive nutrients, are easily digested and absorbed, and have various physiological and health benefits. Moreover, the consumption of plant protein contributes to reduced carbon emissions and promotes resource conservation compared with animal proteins [11]. Among various plant proteins, quinoa protein (QP) emerges as a candidate with significant developmental potential due to its high protein content, balanced essential amino acid profile, and absence of gluten [12]. Prior research on QP has confirmed that the predominant proteins found in quinoa are globulins (37%), albumins (35%), and a relatively small proportion of prolamins (0.5–7.0%) [13]. QP contains a complete amino acid profile, including the essential amino acids required for the growth and maintenance of metabolic activity. In particular, QP exhibits a high abundance of lysine, histidine, and methionine, which are typically the amino acids that are most limited in common cereal proteins (wheat, corn, and rice) [14,15]. The structural and nutritional superiority of QP, highlighted by its rich globulin content and capacity for forming disulfide bonds, underlines its potential as a food ingredient [16].

Current research has focused on enhancing the gel properties of quinoa protein through various modification methods, aiming to broaden its food application spectrum. Treatments such as enzymatic hydrolysis [17], adjustments to ion type and concentration [18], and the combined application of high-intensity ultrasound and transglutaminase [19] have been explored to improve gel strength and water holding capacity (WHC). The strategy of pairing plant proteins with biopolymers, like polysaccharides, to form functional complexes has been recognized as an effective means to enhance plant proteins’ functional properties [20]. Despite the focus on optimizing preparation conditions for quinoa protein gels, their application as egg replacements in bakery products remains an area ripe for exploration, promising to align with the growing trend towards sustainable and plant-based food solutions. 

In this study, a compound QPG system was constructed by combining ultrasound with XG. The microstructure of the gel was observed using a scanning electron microscope, and the changes in its internal structure were explored using disulfide bonds and Fourier transform infrared spectroscopy (FTIR) analysis. The functional properties of the gel were evaluated by analyses of WHC and oil holding capacity (OHC), and finally the physical properties of the bread were evaluated based on its textural properties. The aim of this study was to prepare high-quality quinoa protein gels and to explore the possibility of their application in bread as an egg substitute. This research provides a theoretical foundation for improving the gelling properties of QPG and developing quinoa products with desirable textures.

## 2. Materials and Methods

### 2.1. Materials and Chemicals

Peeled quinoa seeds (LYLM-3) were obtained from the Gansu Academy of Agriculture Science (Lanzhou, China). Potassium bromide (KBr) and 5,5′-dithio-bis-nitrobenzoic acid (DTNB) were purchased from Aladdin Reagent Co. (Shanghai, China). The bread ingredients, high-gluten flour (Dragon fish, Wuxi, China), high-sugar-resistant yeast powder (Wuxi, China), peanut oil (Wuxi, China), powder sugar (Wuxi, China), and egg (Wuxi, China), were purchased from the local supermarket. Other chemicals were purchased from Sinopharm Chemical Reagent Co., Ltd. (Shanghai, China) and were of analytical grade.

### 2.2. Extraction of QP

QP was obtained according to previous studies [21]. Quinoa was peeled, ground, and passed through a 149 μm sieve to generate a fine powder. The quinoa powder was degreased by treating it with n-hexane; this process was repeated four times.

The defatted quinoa flour was using a 1:12 ratio of water at 37 °C and pH 9 for 3 h. The pH was adjusted to 4.5 to precipitate QP. The water extraction and precipitation processes were repeated three times. Finally, the pH was adjusted to 7.0, and the obtained protein was freeze-dried for future experiments. The purity of the QP was 85%.

### 2.3. Ultrasonic Treatment of Quinoa Protein

QPI was dissolved in deionized water to prepare protein dispersion with concentration c (5, 6, 7, 8%, *w*/*v*). The pH of the protein dispersion was adjusted to 7.5 with 1 M NaOH and HCl, and it was continuously stirred and hydrated for 2 h. The 10 mm ultrasonic probe was inserted into the dispersing liquid, the dispersing liquid was treated at 300 W and 20 kHz, and the ultrasonic density was controlled to reach 8 kJ/mL, respectively. Ultrasonic density is a parameter reflecting the relationship between ultrasonic treatment time, power, and sample volume. According to the research of Bot et al. [22], ultrasonic density parameters can be calculated by the following formula:$$\mathrm{Ultrasonic}\;\mathrm{density}\;\left(\mathrm{kJ}/\mathrm{mL}\right)=\frac{W\times T}{V}$$
where *W* represents ultrasonic power, *T* represents ultrasonic time, and *V* represents the volume of the sample. The above three parameters can be controlled simultaneously by controlling the ultrasonic density. The temperature of the sample in the centrifuge tube was controlled by an ice bath during the whole process of the ultrasonic treatment.

### 2.4. Preparation of Gels

The QPI and ultrasonic treatment quinoa protein (UQPI) were dissolved in water, and suspensions of different concentration were prepared. The suspensions were stirred for 3 h and then placed in a refrigerator overnight to allow for hydration. A 0.4% concentration solution of xanthan gum (XG) was prepared and stirred for 6 h before being left for overnight hydration for later use. The XG solution was mixed 1:1 evenly with quinoa protein solution (or UQPI solution) of different concentrations to prepare gel solutions with 5%, 6%, 7%, and 8% quinoa protein (or UQP) concentrations. The gel solutions were heated at 90 °C for 40 min. After being placed in a 4 °C refrigerator for 24 h, the gels were formed.

### 2.5. Measurement of Zeta Potential

The zeta potential of XG-UQPG and XG-QPG was measured using a model ZS zeta potential/nanoparticle size analyzer (Malvern Instruments, Malverns, UK). To ensure the accuracy of the results, the samples were diluted to 0.2 wt% and stirred evenly before measurement. Mean zeta potentials were obtained using this process.

### 2.6. Sodium Dodecyl Sulphate–Polyacrylamide Gel Electrophoresis (SDS-PAGE) Analysis

SDS-PAGE was used to determine the molecular weight distribution of QPG, as previously described [23]. Briefly, protein suspensions (1% *w*/*w*) were prepared in water and centrifuged for 10 min at 10,000× *g*. The supernatants were diluted with 2× sample buffer and deionized water before applying the samples to Bio-Rad gels. Molecular weight markers (10–250 kDa) were obtained from Bio-Rad Laboratories Inc. The protein bands resolved upon electrophoresis were stained with Coomassie Blue Fast Staining Solution (Beyotime, Shanghai, China).

### 2.7. SEM

Small pieces of gel (about 5 × 5 × 5 mm) were removed from the intact gel samples with a razor blade and then immersed in 2.5% glutaraldehyde fixative for 12 h. The sample crystals were rapidly cold-aged in liquid nitrogen (−196 °C), then freeze-dried, and the dried samples were stored in a desiccator [24]. The microstructures of the samples were observed using an electron microscope (SU8100, Hitachi High-tech Co., Tokyo, Japan), sprayed with gold, and accelerated at 3 kV.

### 2.8. Intermolecular Forces and Disulfide Bond

The determination of the intermolecular interaction force of XG-QPG and XG-UQPG was achieved by referring to the method of Zhang et al. [25] with slight modifications. A PBS solution containing 2% SDS, 10 M urea, and 100 mM dithiothreitol was prepared and mixed with dilute protein solutions of different concentrations in equal volumes. The solution was left to stand for 30 min, transferred to a quartz colorimetric dish, and placed in the sample tank of an ultraviolet spectrophotometer. The emission wavelength was set to 600 nm to accurately measure absorbance.

The sulfhydryl and disulfide bonds of the proteins and gels were determined by the reference method [26]. Free sulfhydryl was measured using 0.02 mL of 4 mg/mL 5,5′-dithiobis-2-nitrobenzoic acid (DTNB) and 2.5 mL of 8 mol/L urea solution (prepared with Tris-Gly) added to 0.5 mL of 10 mg/mL sample and reacted for 25 min at 25 °C. The absorbance values were measured at 412 nm. To determine total sulfhydryl, 0.02 mL of β-hydrophobic ethanol and 1.0 mL of 10 mol/L urea solution (prepared by Tris-Gly) were added to 0.2 mL of 10 mg/mL sample, mixed well, and reacted at 25 °C for 1 h. The absorbance was measured at 412 nm. After allowing the reaction to continue for 1 h at 25 °C, the sample was centrifuged and washed twice. The washed precipitate was dissolved with 3.0 mL of 8 mol/L urea solution (prepared with Tris-Gly), and 0.04 mL of DTNB was added. The reaction proceeded at 25 °C for 25 min, followed by the measurement of the absorbance at 412 nm.

### 2.9. FTIR

FTIR analysis of XG-UQPG and XG-QPG was conducted using IS-10 (Nicolet, Waltham, MA, USA). Additionally, the relative proportion of the secondary structural components was estimated by analyzing the amide I region, following a reported protocol [27].

### 2.10. Rheology and Texture

The rheometer (DHR-3, Waters, Milford, CT, USA) equipped with an aluminum parallel plate of 40 mm was used to analyze the rheological properties. Frequency scanning was used to study the gel properties and frequency dependence. Flow sweep measurements were conducted to record the viscosity curves of the emulsions as shear rate changes. All rheometer measurements were carried out at a temperature of 25 °C.

The gel texture parameters were determined using a TA-TX2 plus texture analyzer (SMS, London, UK), as previously described, with certain modifications [28]. A force–time curve was obtained at a crosshead speed of 0.50 mm/s for a 15 mm displacement, with pretest and posttest speeds set at 0.5 and 10 mm/s, respectively. Hardness, springiness, chewiness, gumminess, and resilience were calculated from the resulting TPA curves.

### 2.11. WHC and OHC

To determine WHC, a sample (0.5 g) was weighed in a pre-weighed centrifuge tube. Five milliliters of water was added, swirled for 30 s, left for 30 min, and centrifuged. The excess water was removed, and the samples were weighed again. The WHC was calculated (%) using Equation (1):(1)$$WHC=1-\frac{\left(\;w_{2}-w_{1}\right)}{w_{0}}\times 100$$
where *w*_0_ represents the initial weight of water in the gel (g) and *w*_2_ − *w*_1_ represents the weight of the expelled water (g).

To determine OHC, a sample (0.5 g) was weighed into a pre-weighed centrifuge tube, followed by the addition of 5 mL of oil. The mixture was swirled for 30 s, left for 30 min, and centrifuged. The excess oil was removed, and the tube was weighed again. The OHC was calculated (%) as follows:(2)$$OHC=1-\frac{\left(\;w_{2}-w_{1}\right)}{w_{0}}\times 100$$
where *w*_0_ is the initial weight of oil in the gel (g) and *w*_2_ − *w*_1_ is the weight of the expelled oil (g).

### 2.12. Emulsification (EAI) and Emulsification Stability (ESI)

EAI and ESI were determined using a previously established method [29] with slight modifications. Briefly, 0.05 mL of the emulsion was collected from the bottom of the beaker at 0 and 10 min and diluted with 0.1% *w*/*w* SDS. The optical density at 500 nm was measured using an SDPTOP UV-2400 spectrophotometer (Shimadzu, Guangzhou, China), with the SDS solution as a blank. The EAI and ESI values were calculated using Equations (3) and (4):(3)$$EAI\left(\frac{m^{2}}{g}\right)=\frac{2\times 2.303\times A^{0}\times D_{f}}{c\times \emptyset \times \left(1-\theta \times 10000\right)}$$
(4)$$ESI\left(min\right)=\frac{A^{0}}{A^{0}-A^{10}}\times 10$$
where *A*^0^/*A*^10^ represents the optical density of the diluted emulsion at 0 and 10 min, *D_f_* denotes the dilution multiple (100), *c* is the protein concentration of the gel solution (0.01 g/mL), and *Φ* represents the volume fraction of the oil in the emulsion (0.25). 

### 2.13. Bread Properties

#### 2.13.1. Bread Preparation

The efficacy of gel as an egg substitute was assessed through a comparative bread baking experiment. The formulation of the bread comprised 150 g of high-gluten flour (the protein content is 13.7 g/100 g), 1.5 g of yeast, 18 g of sugar, 1.8 g of salt, 70 g of water, 15 g of peanut oil, and QPG (50 g, control bread was made using egg) at a protein concentration of 6%. The dry ingredients—sugar, salt, and yeast—were initially integrated into the high-gluten flour, followed by the addition of water. This mixture was then stirred in a bread kneader (Shenzhen Shepherd Electric Hardware Co., LTD., Shenzhen, China) at a slow speed, transitioning to a medium speed for 15 min. Subsequently, peanut oil and the gel solution (or equal weight of the egg mixture) were incorporated, with the mixture being stirred at medium speed for an additional 15 min, and then at high speed for 20 min until a smooth dough consistency was achieved. The dough was allowed to rest for 15 min, after which it was portioned, rounded, and allowed to rest again for 15 min before shaping, molding, and placement on a baking tray. The dough was awakened again for 60 min at 30 °C and 75% humidity. Then, the bread was cooled to room temperature after baking at 180 °C for 28 min.

#### 2.13.2. Determination of Texture of Bread

The textural characteristics of the bread were evaluated using a textural analyzer (TA-XT2). We removed the edge of the bread and cut a 4 × 4 × 1.5 cm piece from the center and sides of the bread. For the TPA, we used a 36 mm diameter cylindrical aluminum probe with a velocity of 5 mm S^−1^ before the test and 0.25 mm S^−1^ after the test, where the time interval was 10 s and the deformation amount was 50%. The texture profile analysis (TPA) test was performed by two bite pressure tests.

### 2.14. Statistical Analysis

The experiments were conducted in triplicate, and the means of the datasets were calculated. A one-way analysis of variance (ANOVA) and Duncan’s multiple range test were carried out to compare the means at a significance level of 5% using IBM SPSS statistics 25 (SPSS Inc., Chicago, IL, USA). Graphs were performed using Origin 2023 software.

## 3. Results and Discussion

### 3.1. Effect of Ultrasound Treatment and pH on Quinoa Protein Gels

#### 3.1.1. Molecular Weight Determination

SDS-PAGE was performed to detect variations in the molecular weight resulting from various treatments. XG-QPG (pH 7.0), XG-UQPG (pH 7.0), and XG-UQPG (pH 4.5) contained typical QP subunits in the 100–10 kDa range (Figure 1B). The protein bands resolved by SDS-PAGE revealed that the subunit size of QP remained unchanged after undergoing ultrasonic treatment and the addition of XG. XG-QPG and XG-UQPG showed the main bands of 10, 22, 23, 29, 35, and 50 kDa, with the 29 kDa band being the most abundant. A previous study reported that the major components of QP are 11S globulin and 2S albumin [30]. 11S globulin is represented by bands ranging from 30 to 40 kDa and 50 kDa. 2S albumin is represented by bands at 10 kDa and lower. 11S globulin is similar in structure to soybean 11S globulin and is composed of six pairs of acid-based polypeptides connected by disulfide bonds [15]. Wang et al. showed that ultrasound did not change the subunit structure of mung bean protein; however, it did ultimately improve the properties of mung bean protein gels [31]. In addition, the study showed that 11S globulin is a key component of soy protein in the gel process [32]. Quinoa 11S globulin may have a potential value for applications because its subunit structure is similar to that of soybean 11S globulin.

#### 3.1.2. Zeta Potential

The Zeta potential serves as a crucial quantifiable measure that reflects the degree of mutual repulsion or attraction between particles within a system. A higher absolute value of the Zeta potential corresponds to a greater level of stability within the system [33]. The isoelectric point of QP is 4.5. At this pH, the potential value of XG-UQPG (pH 4.5) sample was close to 0, indicating that the protein was in an unstable state. This instability led to the aggregation of protein particles, as revealed by the SEM diagram. At pH 7.0, the absolute potential of XG-UQPG is highest, indicating that XG-UQPG is more stable. This may be due to the action of ultrasound, which activates the hydrophobic group of the protein, thereby exposing a potential amino acid and resulting in an increase in the absolute potential.

#### 3.1.3. Microstructure

Scanning Electron Microscopy (SEM) images of XG-QPG (pH 7.0), XG-UQPG (pH 7.0), and XG-UQPG (pH 4.5) are shown in Figure 2A–L. All the gels exhibited a lamellar structure at the macro level. XG-QPG (pH 7.0) showed a network of different pore sizes, and unbound globular proteins were observed in the network at 5000× (Figure 2A–D). Conversely, XG-UQPG (pH 7.0) demonstrated a looser structure with uniform voids (Figure 2E–H). These findings suggest that ultrasonic treatment can improve the homogeneity of the gel network. Such structural enhancements at the macro level contributed to improvements in WHC. The variations in the gel network may be attributed to the ultrasonic treatment, which reduced the particle size and unfolded the structure of QP, exposing more binding sites for XG and proteins [34]. Furthermore, this treatment exposed active groups that promoted the formation of intermolecular disulfide bonds and hydrophobic interactions [28]. 

In comparison, XG-UQPI (pH 4.5) exhibited a denser lamellar structure than XG-UQPG (pH 7.0) (Figure 2I–L). At a pH of 4.5, corresponding to the isoelectric point (Ip) of quinoa protein, the protein molecules lack sufficient charge to repel each other, leading to weakened intermolecular forces, thus facilitating particle condensation. This condition also diminishes the electrostatic interaction and aggregation between quinoa protein and XG, resulting in smaller pores and more robust protein chains at pH 4.5. These results align with the observations made by M.C. Cortez-Trejo [35] in their investigation of mixed gelation involving bean protein and XG. Their study revealed that at low pH levels, the formation of compact structures was due to electrostatic interactions between the protein and polysaccharide. Furthermore, the study of vana M. Geremias-Andrade also confirmed the spherical aggregator structure with different sizes and pores in the gel, in which the fine filamentous connector was XG [36].

#### 3.1.4. Intermolecular Forces

Figure 3A–D show the protein absorbance of the gels in several solvents related to the disulfide, hydrophobic, and hydrogen bond interactions involved in gel formation [37,38,39]. When SDS, urea, and DTT were added to all the dispersions, the absorbance of the dispersions increased, implying that the stability of the gels was maintained through the combined contributions of hydrophobic interactions, hydrogen bonds, and disulfide bonds.

At pH 7, hydrogen and disulfide bonds were the main interaction in XG-QPG (pH 7.0). This may be attributed to the hydrophobic groups of the QPI molecules being wrapped inside without ultrasonic treatment, and the connection between the QPI and XG molecules was not sufficient. SEM images also confirmed that individual proteins were isolated outside the protein chain and not attached to the gel network structure.

XG-UQPG (pH 7.0) displayed increased absorbance with DTT, followed by the SDS group, which indicated that the main forces were disulfide bonds and hydrophobic interactions. These results suggested that the disulfide bonds play a major role in the composite gel, consistent with the results for sulfhydryl bonds. Because of XG-QPG (pH 7.0) and XG-UQPG (pH 7.0), ultrasound treatment can unfold the structure of protein molecules and disrupt non-covalent interactions, thereby reducing hydrophobic interactions and promoting the formation of intermolecular disulfide bonds. 

In the case of pH 4.5, the addition of SDS notably increased absorbance in XG-UQPG (pH 4.5), indicating the crucial role of hydrophobicity. This pH level corresponds to the isoelectric point of QPI, where proteins are partially denatured and hydrophobic groups are exposed, leading to the formation of large aggregates. 

#### 3.1.5. Disulfide Bond

Table 1 describes the measurements of free sulfhydryl, total sulfhydryl, and disulfide bonds across all the samples. Notably, the total sulfhydryl group and disulfide bond of XG-UQPG (pH 7.0) reached the maximum value (166.12 μmol/g and 61.58 μmol/g) at a protein concentration of 6%, higher than XG-QPG (pH 7.0). XG-UQPG (pH 4.5) exhibited a high content of free sulfhydryl and a low content of total sulfhydryl and disulfide bonds, which is consistent with the results of the intermolecular forces. Overall, the free sulfhydryl group content of XG-UQPG (pH 4.5) was higher than that of the other samples because a few proteins in the gel did not form gel networks as protein chains. However, protein monomers were observed between the gel networks. This pattern indicated that ultrasonic treatment, by exposing hydrophobic protein groups, facilitated the interaction between XG and QPI, thereby enhancing disulfide bond formation within the gel, particularly at pH 7.0. Previous studies have shown that SH-SS present in gluten subunits undergoes exchange reactions, forming intermolecular disulfide bonds during dough mixing and eventually forming polymer networks along extended directions [40]. The high disulfide bond content in the gels may promote the formation of a gel network between gluten proteins, thereby enhancing bread elasticity.

#### 3.1.6. Rheology

Figure 4A–D display the frequency-dependent behavior of the storage modulus (G′) and the loss modulus (G″) for our gel samples. With increasing frequency, G′ increased and was larger than G″, showing obvious elastic hydrogel behavior. Among them, XG-UQPG (pH 7.0) exhibited the largest value of G′, reaching 8 × 10^-5^ MPa. This enhancement can be attributed to ultrasonic treatment, which unfolds the hydrophobic structure of protein molecules and release-free sulfhydryl groups, thus promoting the connection of XG and QPI molecules to form a more stable chain group structure. At pH 4.5, near the protein’s isoelectric point, protein aggregation and insufficient QPI and XG binding lead to heterogeneity, mirroring the findings from Yang’s research on heat-induced gels and the ultrasonic treatment of soy protein [41]. The observation that G′ displayed frequency dependence, whereas G″ remained relatively unaffected by frequency changes, aligns with the literature, indicating that food gels, as viscoelastic materials, typically show such behavior [42].

Figure 4F–H show the changes in the viscosity and shear stress of the gels with the shear rate. The shear stress of all the samples increased with an increasing shear rate, while the viscosity decreased with an increasing shear rate or shear stress. Throughout the entire range of shear rates, all samples displayed pseudoplastic behavior, also known as shear thinning, in which the viscosity decreased as the shear rate increased. Other plant protein gels exhibit the same shear thinning behavior [43,44]. At the same pH, XG-UQPG (pH 7.0) exhibits greater viscosity than XG-QPG (pH 7.0). For the same strain, XG-UQPG (pH 7.0) requires greater stress, which indicates that ultrasound improves the consistency of the gel solution. These changes are likely due to ultrasonic treatment altering the protein’s natural state, affecting properties such as solubility, particle size, and microstructure, and consequently modifying the flow characteristics of the XG-UQPG dispersions. Specifically, XG-UQPG (pH 7.0) showed increased viscosity and shear stress compared to XG-UQPG (pH 4.5), indicating a higher consistency at neutral pH.

#### 3.1.7. Texture

The texture profile analysis (TPA) assessed the hardness, adhesiveness, springiness, and resilience of the gels, as summarized in Table 2. Among these, hardness is particularly significant because it indicates the maximum force during the initial deformation cycle and reflects the firmness of the gel [45].

The hardness of XG-UQPG (pH 7.0) and XG-QPG (pH 7.0) increased significantly as the protein concentration increased and was less than 40g. Le and Turgeon found that the primary factor contributing to the observed increase in strength during a penetration test is the rise in dense regions within the gel network [46]. SEM imaging revealed that at pH 7.0, the network structure of XG-UQPG (pH 7.0) and XG-QPG (pH 7.0) was comparably looser, characterized by larger pores, contributing to the relatively modest hardness observed at this pH.

At pH 4.5, XG-UQPG (pH 4.5) reached its maximum hardness at a QP concentration of 6% (142 g). The hydrophobic interaction at pH 4.5 is the main force, and the proteins gather together to make the structure dense and strong. A protein concentration of 6% was considered to make the optimal concentration for XG and QPI junctions. The hardness values for XG-QPG and XG-UQPG were in the range of 15–142 g, which was comparable to the findings of Micaela Galante’s study. Galante reported hardness values of up to 164 g for QPHG gels formulated with a higher concentration of bean protein isolate and carbohydrate (10% and 4%, respectively) and denaturation temperatures of 95 °C [17]. 

Springiness, another measure related to gel network density, was also influenced by ultrasound treatment and pH [47]. Post-ultrasonic treatment, XG-UQPG (pH 7.0) exhibited significantly higher springiness than XG-QPG (pH 7.0). Elasticity is linked to the disruption of the internal structure of the gel, suggesting that ultrasound enhanced the anti-deformation ability, resulting in the formation of a highly elastic gel. However, at pH 4.5, XG-UQPG (pH 4.5) has slightly higher elasticity than XG-UQPG (pH 7.0). This finding implied that at a low pH, the elasticity of the gel is higher, possibly because of the strong binding of the protein molecules, and the protein chains promote the deformation resistance of the gel network.

Changes in chewiness and resilience followed the same trend as the changes in elasticity. This indicates that both ultrasound and pH can enhance the chewiness and resilience of the gels, with pH having the greatest impact.

#### 3.1.8. FTIR

FTIR was used to assess the secondary structure of the samples by analyzing the conventional amide I band. This band was subdivided into four regions indicative of different secondary structures: 1600 to 1640 cm^−1^ for β-sheet, 1640 to 1650 cm^−1^ for random coil, 1650 to 1660 cm^−1^ for α-helix, and 1660 to 1700 cm^−1^ for β-turn, as detailed in Figure 3E–H. 

In the case of XG-QPG (pH 7.0), the predominance of the α-helix structure was observed, with lower proportions of β-sheet and β-turn. In contrast, for XG-UQPG at the same pH, there was a noticeable reduction in α-helix and random coil, while there was a significant increase in β-sheet and β-turn. These findings indicate that ultrasonic treatment leads to a more orderly protein conformation in XG-UQPG. This phenomenon may be attributed to the partial unfolding of the protein structure induced by the sheer force and cavity effects caused by ultrasonication. Consequently, the local amino acid sequences and interactions between protein molecules were disrupted, leading to the transformation of insoluble aggregates into soluble aggregates [34]. The aggregation of QPI may be attributed to the formation of β-sheet.

The secondary structure of XG-UQPG (pH 4.5) protein showed a similar trend to XG-UQPG (pH 7.0), albeit with a higher content of β-sheet. Previous research has indicated a correlation between the presence of β-sheet and the mechanical robustness of the gel. Specifically, an increase in the β-sheet content was associated with an enhancement of gel strength, similar to the texture results. The gel strength at pH 4.5 was greater than the strength at pH 7.0.

#### 3.1.9. WHC and OHC

WHC and OHC are pivotal attributes of food gels, often correlating directly with their microstructural properties [48]. In this study, at pH 7.0, the WHC and OHC of XG-UQPG (pH 7.0) were higher than the value of XG-QPG (pH 7.0). The WHC and OHC of XG-UQPG (pH 4.5) were lower than the value of XG-UQPG (pH 7.0). These results suggested that XG-UQPG (pH 7.0) exhibits stronger WHC and OHC, especially at protein concentration of 6% (Figure 5A,B). These results further indicate that ultrasound substantially improves the OHC and WHC of gel, particularly at pH 7.0. This may be related to the porosity of XG-UQPG (pH 7.0); the larger the pore size, the more water molecules are stored; thus, a higher WHC is obtained. Simultaneously, the gels exhibited thinner structures and weaker electrostatic associations, allowing both proteins and XG to retain water at pH 7.0.

In contrast, the gels had a denser and more compact microstructure, with minor biopolymer content available for water retention at pH 4.5. At this pH, proteins were at their isoelectric points, resulting in protein aggregation and reduced connections between the protein chains and XG. Consequently, the available space within the gel network decreased, leading to a corresponding decrease in the amount of retained water and oil. Furthermore, the WHC and OHC of the gel at both pH values initially increased and then decreased with an increasing protein concentration. This could be partly attributed to the fact that the protein was most closely connected to XG at a protein concentration of 6%. Given the crucial role of hydrocolloids in improving WHC in bread, influencing water distribution, and affecting starch retrogradation [49], the findings imply that XG-UQPG at pH 7.0 could be a viable egg substitute in bread formulations, especially considering its superior WHC.

#### 3.1.10. Emulsification and Emulsification Stability

EAI and ESI are frequently used to assess emulsifying properties. EAI measures the ability of proteins to be adsorbed at the oil/water interface during emulsion preparation. ESI reflects their capacity to maintain stability [29,50]. The EAI of XG-UQPG (88.28 m^2^/g) and XG-QPG (19.65 m^2^/g) was the highest at a protein concentration of 6% at pH 7.0. Additionally, the ESI of XG-UQPG (pH 7.0) of 80.05min was significantly higher than that of XG-QPG (pH 7.0) of 24.76 min, indicating that ultrasound treatment can significantly promote the EAI and ESI of gels. This discrepancy highlighted the enhanced emulsifying capability and stability conferred by ultrasonic treatment. The ultrasonic treatment disrupted the structure of the protein molecules, exposing the internal hydrophobic groups. This exposure of hydrophobic groups facilitated the interaction among proteins and oil, causing the formation of a balanced water/oil interface, thereby improving the emulsification activity and stability.

At the same concentration, the EAI and ESI of XG-UQPG (pH 7.0) were higher than the values at pH 4.5. These findings indicate that at pH 7.0, the composite gel exhibited stronger emulsification and higher emulsion stability, especially at a protein concentration of 6%. The pH affected the solubility and aggregation of the protein molecules, with XG binding to UQPI molecules being more effective at a neutral pH, as mentioned earlier. The highest ESI and EAI at a protein concentration of 6% may be attribute to the ultrasound treatment, which exposed hydrophobic groups and allowed for a better balance between protein structure unfolding and cross-linking in XG-UQPG (pH 7.0), resulting in superior emulsification stability and activity.

Meanwhile, XG-UQPG (pH 7.0) with strong emulsification properties can stabilize dough by interacting with thermodynamically unstable gluten proteins. The addition of XG-UQPG (pH 7.0) during mixing can enhance the strength and ductility of the dough. Furthermore, throughout fermentation, it aids in air retention and mitigates dough collapse, underscoring its potential utility in enhancing bread quality [51].

### 3.2. Effect of Complex Gels on Physical Properties of Bread

#### 3.2.1. Texture

A significant correlation has been identified between the textural properties of bread and its sensory evaluation, indicating that these textural attributes play a crucial role in shaping the consumer’s sensory experience and overall satisfaction. It is generally observed that consumers prefer bread that is soft and moist with a certain degree of elasticity. This preference is due to the fact that bread with a soft, moist texture and adequate elasticity provides a more pleasing sensory experience. In contrast, bread characterized by hardness and low chewability is perceived as having a tougher taste and lacks the desired softness and refreshing quality. Conversely, bread that exhibits higher levels of elasticity and resilience is perceived as softer, offering a refreshing and less sticky mouthfeel [52].

Egg bread demonstrated the highest level of hardness, followed in order by XG-UQPG (pH 7.0), XG-UQPG (pH 4.5), and XG-QPG (pH 7.0), with XG-QPG being the softest and most pliable (Table 3). Furthermore, in terms of springiness and resilience, XG-UQPG (pH 7.0) exhibited superior quality, which is in agreement with the findings from previous research. This enhancement of bread elasticity can be attributed to an increase in the disulfide bond content within XG-UQPG (pH 7.0), which strengthens the gel network structure. Additionally, the observed improvements in WHC and EAI for XG-UQPG (pH 7.0) likely contribute to a higher moisture content in the bread, enhancing its sensory attributes. Consequently, these results suggest that XG-UQPG (pH 7.0) bread, with its soft and elastic texture, is more appealing to consumers.

This detailed analysis underscores the importance of textural parameters in the sensory evaluation of bread by consumers. Understanding the relationship between specific textural characteristics and consumer preferences can guide product development and improvement, ensuring that bread varieties meet or exceed consumer expectations for texture and overall sensory experience.

#### 3.2.2. Appearance

Figure 6A–D show bread made from egg, XG-QPG (pH 7.0), XG-UQPG (pH 7.0), and XG-UQPG (pH 4.5) at a protein concentration of 6%. Visually, XG-UQPG (pH 7.0) appeared to be well fermented, with a shape similar to commercially available bread. However, bread prepared using XG-QPG (pH 7.0) and XG-UQPG (pH 4.5) displayed pits on both sides (As shown by the red line in Figure 6). These physical characteristics were influenced by the behavior of globulins within the gel during baking, where protein development and aggregation fostered the creation of three-dimensional networks, transforming the batter into bread [53]. Free sulfhydryl groups in proteins can form intermolecular disulfide bonds and enhance the filter cake structure. Notably, the disulfide bond content in XG-UQPG (pH 4.5) was lower than in XG-UQPG (pH 7.0), which could account for the observed depressions in the bread’s surface. Among the four samples, XG-QPG and XG-UQPG (pH 7.0) displayed dense and uniform pores, whereas egg bread (Figure 6A) had large pores. In conclusion, ultrasonic treatment significantly improves the properties of gel when combining XG and QPI, making it a viable substitute for eggs in bread and to enhance the properties of bread.

## 4. Conclusions

This study explored how ultrasound treatment could improve quinoa protein gels in order to use them as egg substitutes in bread. We found that this treatment changes the gel’s structure and how it behaves at different acidity levels. Specifically, it helped proteins organize themselves better and form strong bonds, essential for creating a good gel. Depending on the pH, the gels showed different textures, from loose with even pores (pH 7.0) to dense with small pores (pH 4.5). Applying ultrasound also enhanced the gel’s practical qualities, like holding more water and oil, and making better emulsions, making it a promising egg alternative in bread. Moreover, considering the antioxidant properties of quinoa protein, future research could delve into the extended storage qualities of these gels and their application in bread, potentially expanding the scope of egg-free baking solutions. This research not only contributes to the field of food science by enhancing the versatility of quinoa protein gels but also opens up new pathways for the development of sustainable and health-conscious food products.

## Figures and Tables

**Figure 1 foods-13-01271-f001:**
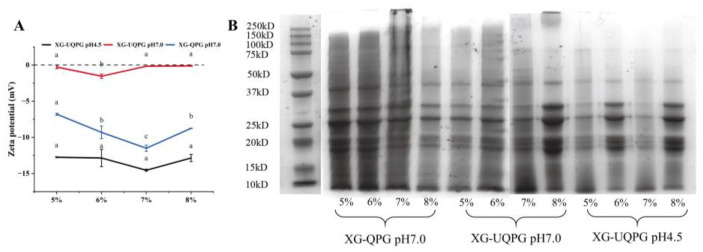
Zeta potential and SDS-PAGE analyses. (**A**) Zeta potential of XG-QPG (pH 7.0), XG-UQPG (pH 7.0), and XG-UQPG (pH 4.5). (**B**) SDS-PAGE of XG-QPG (pH 7.0), XG-UQPG (pH 7.0), and XG-UQPG (pH 4.5). a–c indicate significant differences between different samples.

**Figure 2 foods-13-01271-f002:**
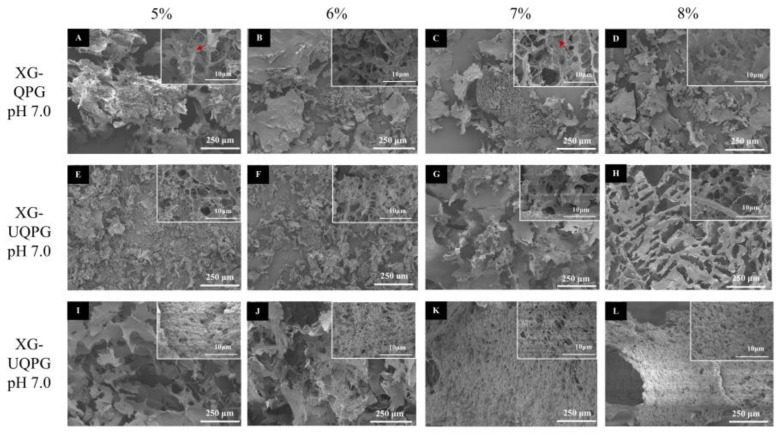
SEN analyses. (**A**–**D**) SEM micrographs of XG-QPG at pH 7.0 and 100×. (**E**–**H**) SEM micrographs of XG-UQPG at pH 7.0 and 100×. (**I**–**L**) SEM micrographs of XG-UQPG at pH 4.5 and 100×. The added protein concentrations are 5%, 6%, 7%, and 8% from left to right.

**Figure 3 foods-13-01271-f003:**
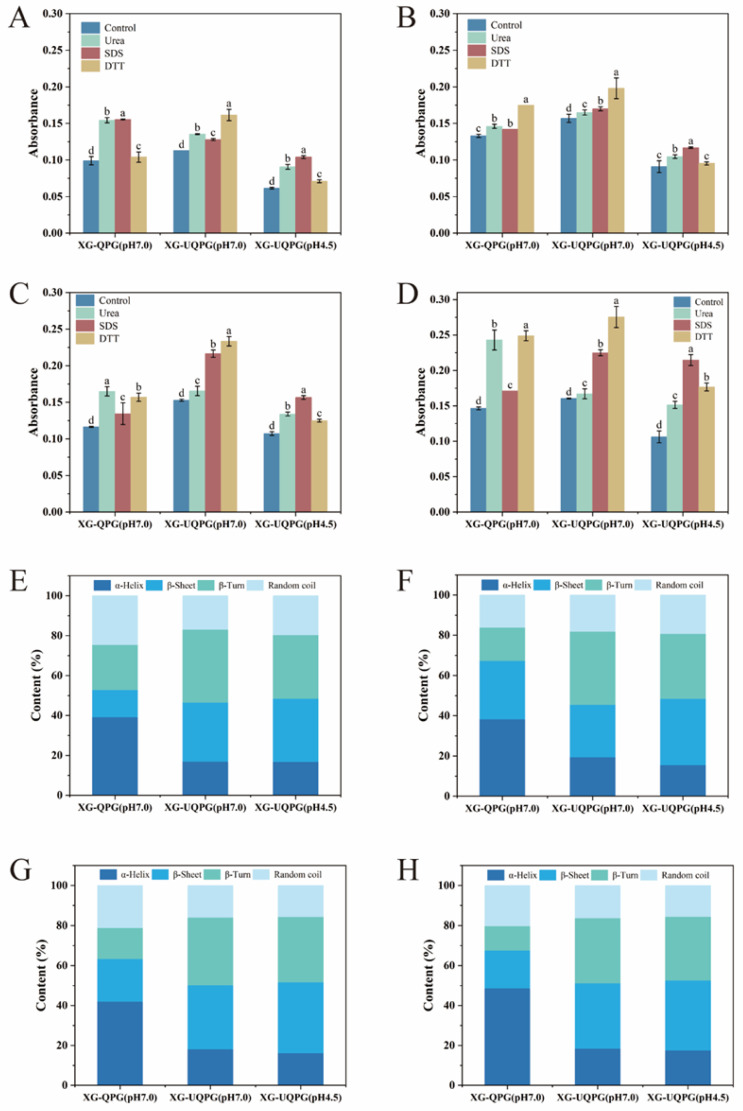
Analyses of QPGs. (**A**–**D**) Intermolecular interaction of XG-QPG (pH 7.0), XG-UQPG (pH 7.0), and XG-UQPG (pH 4.5) at protein concentrations of 5%, 6%, 7%, and 8%. (**E**–**H**) Protein secondary structure of XG-QPG (pH 7.0), XG-UQPG (pH 7.0), and XG-UQPG (pH 4.5) at protein concentrations of 5%, 6%, 7%, and 8%. The letters a–d indicate significant differences between different samples.

**Figure 4 foods-13-01271-f004:**
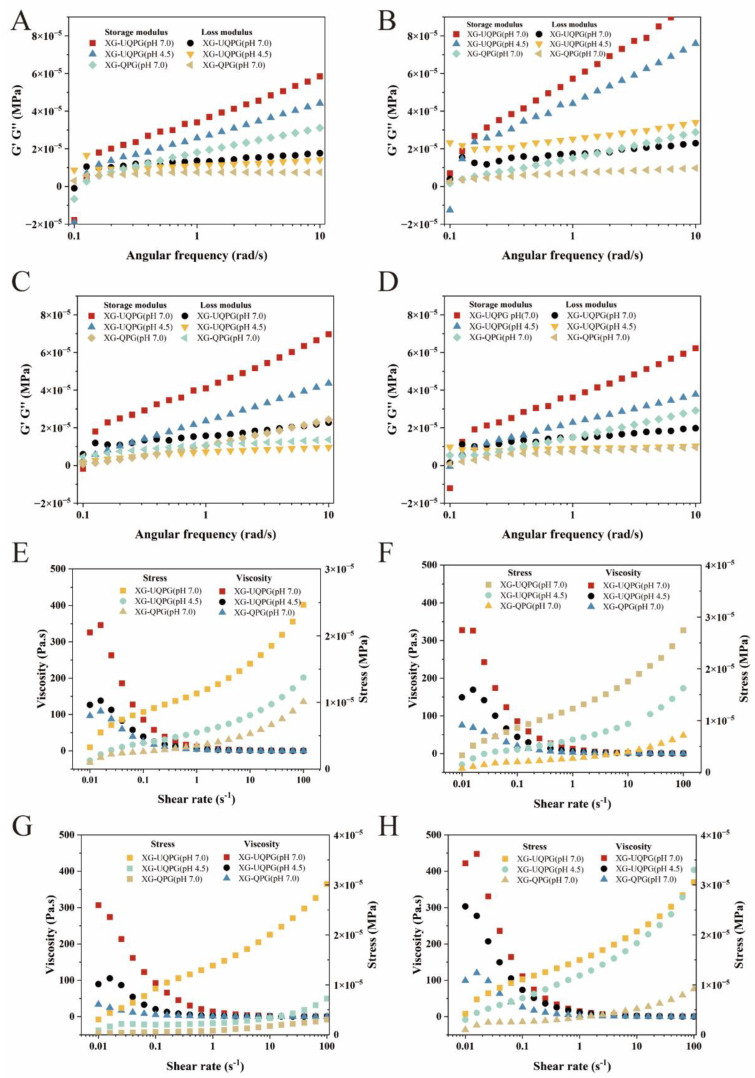
Frequency and shear scan data. (**A**–**D**) Frequency scans of XG-QPG (pH 7.0), XG-UQPG (pH 7.0), and XG-UQPG (pH 4.5) at protein concentrations of 5%, 6%, 7%, and 8%. (**E**–**H**) Shear scans of XG-QPG (pH 7.0), XG-UQPG (pH 7.0), and XG-UQPG (pH 4.5) at protein concentrations of 5%, 6%, 7%, and 8%.

**Figure 5 foods-13-01271-f005:**
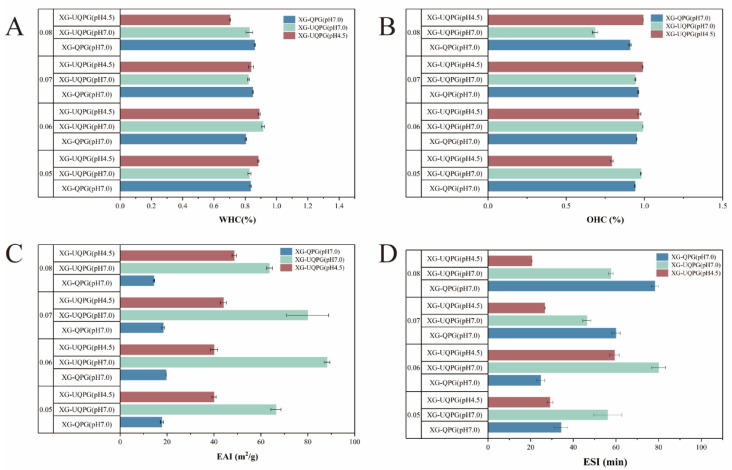
Retention and stability analyses. (**A**,**B**) WHC and OHC of XG-QPG (pH 7.0), XG-UQPG (pH7.0), and XG-UQPG (pH 4.5) at protein concentrations of 5%, 6%, 7%, and 8. (**C**,**D**) EAI and ESI of XG-QPG (pH 7.0), XG-UQPG (pH 7.0), and XG-UQPG (pH 4.5) at protein concentrations of 5%, 6%, 7%, and 8%.

**Figure 6 foods-13-01271-f006:**
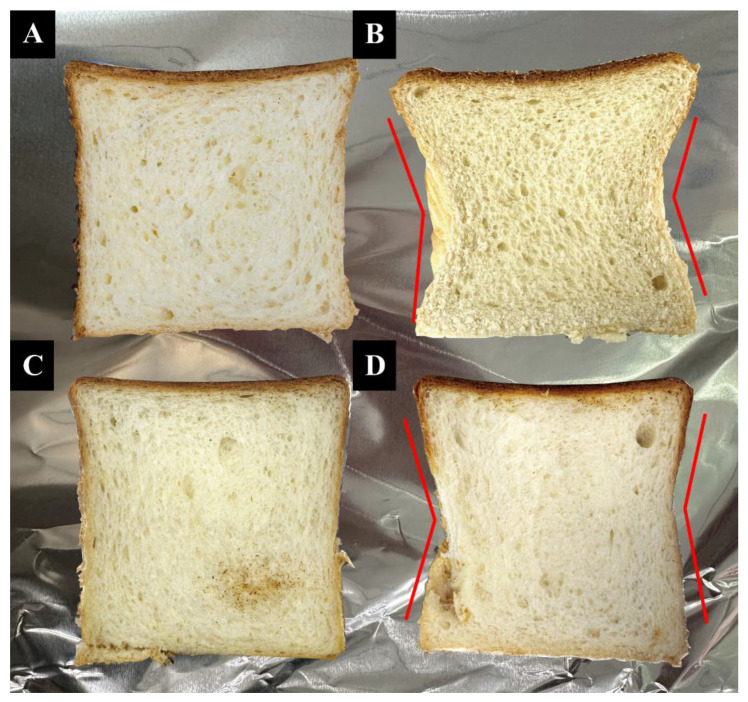
Appearance of breads. (**A**) Egg bread. (**B**–**D**) Bread made with XG-QPI (pH 7.0) (**B**), XG-UQPI (pH 7.0) (**C**), and XG-UQPI (pH 4.5) (**D**) instead of egg at a concentration of 6%.

**Table 1 foods-13-01271-t001:** Total sulfhydryl, free sulfhydryl, and disulfide bonds of XG-QPG (pH 7.0), XG-UQPG (pH 7.0), and XG-UQPG (pH 4.5).

Protein Concentration		Total Sulfhydryl Group	Free Sulfhydryl	Disulfide Bond
		(umol/g)	(umol/g)	(umol/g)
**5%**	**XG-QPG (pH 7.0)**	110.91 ± 4.79 ^c^	47.90 ± 1.13 ^b^	31.51 ± 1.83 ^def^
	**XG-UQPG (pH 7.0)**	115.21 ± 0.92 ^bc^	19.34 ± 0.69 ^e^	47.93 ± 0.23 ^ab^
	**XG-UQPG (pH 4.5)**	106.27 ± 2.82 ^c^	74.21 ±1.41 ^a^	16.03 ± 0.70 ^g^
**6%**	**XG-QPG (pH 7.0)**	124.63 ± 2.81 ^b^	48.30 ± 0.60 ^b^	38.17 ± 1.70 ^cd^
	**XG-UQPG (pH 7.0)**	155.12 ± 1.58 ^a^	31.95 ± 0.30 ^d^	61.58 ± 0.64 ^a^
	**XG-UQPG (pH 4.5)**	112.05 ± 0.36 ^bc^	43.66 ± 6.47 ^bc^	34.20 ± 3.42 ^de^
**7%**	**XG-QPG (pH 7.0)**	82.74 ± 5.49 ^d^	33.95 ± 0.01 ^cd^	24.40 ± 2.75 ^f^
	**XG-UQPG (pH 7.0)**	107.24 ± 2.01 ^c^	35.64 ± 0.47 ^cd^	35.80 ± 1.24 ^de^
	**XG-UQPG (pH 4.5)**	105.37 ± 2.40 ^c^	80.27 ± 8.65 ^a^	12.55 ± 3.13 ^g^
**8%**	**XG-QPG (pH 7.0)**	103.55 ± 1.91 ^c^	43.17 ± 0.13 ^bc^	30.19 ± 1.02 ^ef^
	**XG-UQPG (pH 7.0)**	109.61 ± 8.17 ^c^	19.65 ± 1.27 ^e^	44.98 ± 3.45 ^bc^
	**XG-UQPG (pH 4.5)**	83.08 ± 3.97 ^d^	52.20 ± 0.46 ^b^	15.44 ± 0.70 ^g^

Different letters in the same column indicated that the analyzed samples are significantly different (*p* < 0.05).

**Table 2 foods-13-01271-t002:** Texture data of XG-QPG (pH 7.0), XG-UQPG (pH 7.0), and XG-UQPG (pH 4.5).

	Samples	Hardness	Springiness	Chewiness	Resilience
		gf		gf	
**5%**	**XG-QPG (pH 7.0)**	15.61 ± 0.11 ^d^	0.46 ± 0.01 ^bcde^	4.37 ± 0.15 ^d^	0.032 ± 0.006 ^c^
	**XG-UQPG (pH 7.0)**	17.83 ± 2.30 ^d^	0.51 ± 0.04 ^abc^	6.63 ± 0.00 ^d^	0.036 ± 0.003 ^c^
	**XG-UQPG (pH 4.5)**	79.32 ± 13.56 ^c^	0.52 ± 0.01 ^abc^	27.07 ± 4.50 ^c^	0.133 ± 0.000 ^a^
**6%**	**XG-QPG (pH 7.0)**	17.61 ± 0.76 ^d^	0.48 ± 0.01 ^bcd^	5.33 ± 0.53 ^d^	0.035 ± 0.000 ^c^
	**XG-UQPG (pH 7.0)**	20.88 ± 1.94 ^d^	0.54 ± 0.02 ^abc^	7.44 ± 0.34 ^d^	0.037 ± 0.002 ^c^
	**XG-UQPG (pH 4.5)**	142.68 ± 11.71 ^a^	0.62 ± 0.02 ^a^	58.68 ± 8.63 ^a^	0.111 ± 0.003 ^ab^
**7%**	**XG-QPG (pH 7.0)**	21.10 ± 2.18 ^d^	0.37 ± 0.10 ^de^	4.07 ± 1.11 ^d^	0.033 ± 0.011 ^c^
	**XG-UQPG (pH 7.0)**	22.69 ± 1.03 ^d^	0.44 ± 0.01 ^cde^	5.84 ± 0.19 ^d^	0.037 ± 0.001 ^c^
	**XG-UQPG (pH 4.5)**	105.47 ± 13.30 ^b^	0.59 ± 0.01 ^a^	41.40 ± 5.35 ^b^	0.094 ± 0.011 ^b^
**8%**	**XG-QPG (pH 7.0)**	23.91 ± 0.59 ^d^	0.35 ± 0.01 ^e^	3.35 ± 0.44 ^b^	0.032 ± 0.003 ^c^
	**XG-UQPG (pH 7.0)**	32.02 ± 1.73 ^d^	0.57 ± 0.02 ^ab^	10.59 ± 0.60 ^d^	0.035 ± 0.003 ^c^
	**XG-UQPG (pH 4.5)**	109.87 ± 5.41 ^b^	0.60 ± 0.01 ^a^	47.11 ± 1.25 ^b^	0.131 ± 0.018 ^a^

Different letters in the same column indicated that the analyzed samples are significantly different (*p* < 0.05).

**Table 3 foods-13-01271-t003:** Hardness, springiness, chewiness, gumminess, and resilience of bread samples.

Samples	Hardness	Springiness	Chewiness	Resilience
	(g)	(g)	(g)	
**Egg bread**	234.91 ± 1.46 ^a^	0.88 ± 0.0037 ^ab^	165.44 ± 0.50 ^a^	0.18 ± 0.0064 ^b^
**XG-UQPG (pH 4.5)**	119.48 ± 2.16 ^c^	0.83 ± 0.0305 ^b^	76.99 ± 1.70 ^c^	0.17 ± 0.0032 ^b^
**XG-UQPG (pH 7.0)**	137.67 ± 3.11 ^b^	0.91 ± 0.0096 ^a^	97.77 ± 1.80 ^b^	0.20 ± 0.0032 ^a^
**XG-QPG (pH 7.0)**	108.08 ± 2.11 ^d^	0.83 ± 0.0043 ^b^	64.91 ± 2.17 ^d^	0.17 ± 0.0019 ^b^

Different letters in the same column indicated that the analyzed samples are significantly different (*p* < 0.05).

## Data Availability

The datasets generated for this study are available on request to the corresponding author. The data are not publicly available due to privacy restrictions.

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
