# Peer review of "The Preparation and Characterization of Quinoa Protein Gels and Application in Eggless Bread"

_foods, 2024, doi:10.3390/foods13081271_

Round 1

Reviewer 1 Report

Comments and Suggestions for Authors

The objective of the present study was to investigate the functional properties of quinoa protein gels prepared by using xanthan gum and ultrasound, as well as to investigate the impact of obtained gels on the quality of bread. The topic of the study is within the scope of journal; however, there are a number of issues that has to be improved in order to get the findings more easily understandable for the reader.

Abstract: In its current form, it looks confusing. Avoid listing values in the abstract because it unnecessarily burdens the text and is difficult to follow. In the abstract section, please provide a clearer insight into the aim and highlight the most important concrete results.

Lines 34-35: such as adding  eggs to supplement egg white protein? Unclear.

Line 36: 3] et al..? Technical error.

The introduction part should be better organized. The introduction consists of random disjointed parts and does not have a good flow. I would suggest the authors give more background information in the introduction in connection with previous research on the modification of quinoa protein gels.

The aim of the study should be expressed in more concise manner incorporating all aspects of the study. Reformulate it.

Material and methods section: It needs to be improved. The methodological approach is not clear enough to fully understand the experimental design and results. From the described methodology, it is not evident how the samples XG-QPG (pH 7.0), XG-UQPG (pH 196 7.0) and XG-UQPG (pH 4.5) were prepared. Moreover, some parts of the method section are written as a protocol; please change it in accordance with the methodology of writing a scientific paper.

Line 102: UQP?

Ultrasound treatment is not described in the method section.

Line 126: reference method? Specify the method source.

EAI and ESI- explain the abbreviations

2.12 Bread Properties - Please provide more details about methodology used for baking procedure (formulation) and evaluation of bread quality.

Line 207: It would be useful to specify which properties.

Results and discussion (general remarks): Please avoid the unnecessary description of the results of each tested sample by ranking them from highest to lowest, as well as inserting the result value because it is clearly visible from the tables or figures and unnecessarily burdens the text. Try to simplify the description of the result by focusing on the influence of ultrasound or pH on the individual properties of gels. The authors have done comprehensive work to characterize gels and manuscript contains a lot of results, but my main complaint is that the description is too long without concrete discussion. Moreover, some statements have to be supported by references. It can easily happen that the reader gets lost while reading the manuscript because it is difficult to follow the text. I kindly suggest to the authors that they try to better summarize the obtained results, unite them, and highlight important points.

Conclusion: Please improve this section by restating the research problem and summarizing the most relevant findings of the paper without stating the values of the examined results.

Author Response

Dear reviewer,

Thank you very much for your kind letter, along with the constructive comments of the editor and reviewers concerning our manuscript (foods-2926222). We have thoroughly considered all the comments and substantially revised our manuscript (modifications are highlighted in yellow).

The point-to-point answers and explanations for all revisions were listed in a separate paper following this letter.

We have tried our best to address all the concerns raised by the reviewers. We hope, with these modifications and improvements based on the comments of the editor and reviewers, the quality of our manuscript would meet the publication standard of Foods.

Yours sincerely,

Li Wang

___________________________

Dr Li Wang

State Key Laboratory of Food Science and Technology

National Engineering Laboratory for Cereal Fermentation Technology

School of Food Science and Technology

Jiangnan University.

Wuxi 214122

  1. R. China

Response to reviewer:

Abstract Revision:

A: Thanks for your suggestion. We have revised the abstract to remove the listing of specific values and focused on clearly stating the aim of our study and highlighting the key findings. The revised abstract now provides a concise overview of our research objectives, methodology, and the significant outcomes.

Lines 34-35:

A: Thanks for your suggestion. I have changed the "egg white protein" to "egg".(Line 32)

Line 36:

A: Thanks for your suggestion. The citation error has been corrected to accurately reflect the referenced study.

Introduction:

A: Thanks for your suggestion. We have restructured the introduction to offer a coherent flow.

Study Aim:

A: Thanks for your suggestion. The aim of the study has been reformulated for greater clarity and succinctness. This revised formulation better guides the reader through our research objectives.

Materials and Methods:

A: Thanks for your suggestion. We have revised the Materials and Methods section for clarity. The preparation of the samples (XG-QPG, XG-UQPG at pH 7.0, and XG-UQPG at pH 4.5) is now described in detail, following the standard scientific paper methodology. The ultrasound treatment methodology is clearly described, ensuring the experimental design is fully transparent.

Line 102 (UQP?)

A: Thanks for your suggestion.The meaning of UQP has been clarified within the text.(Line 125)

Line 126: The reference method is now specified, including the source of the method. EAI and ESI abbreviations are explained at their first mention in the text.

A: Thanks for your suggestion. References have been added in 2.6 and the full names of EAI and ESI have been added in 2.12.

Bread Properties (2.12): Bread Properties - Please provide more details about methodology used for baking procedure (formulation) and evaluation of bread quality.

A: Thanks for your suggestion. We have expanded the section on bread properties, detailing the methodology used for the baking procedure and evaluation of bread quality, to ensure replicability and clarity. (2.13)

Line 207: It would be useful to specify which properties.

A: Thanks for your suggestion. We added a description of the useful indicators to the SDS-PAGE. (Line 255)

Results and Discussion:

A: Thanks for your suggestion. We have simplified the description of results, focusing on the influence of ultrasound and pH on the gel properties and decreased enumerating all the values.

The discussion now emphasizes the main findings, supported by references, and avoids lengthy descriptions of individual results.

Conclusion:

A: Thanks for your suggestion. The conclusion has been improved to succinctly restate the research problem and summarize the most relevant findings without listing specific values.

Reviewer 2 Report

Comments and Suggestions for Authors

Dear Authors,

The reviewed manuscript is an interesting study of the possibility of modifying the bread recipe to replace raw materials of animal origin and allergenic raw materials of plant origin. Despite the interesting topic and quite extensive research part, I see a number of shortcomings both in the planned experiment and in the preparation of the manuscript itself.

I provide comments below:

Manuscript ID: foods-2926222

Title: The preparation, characterization of quinoa protein gels and application in eggless bread

general comments:

The introduction of the work is concise and quite well written, but additions should be made and the wording of the aim of the study should be significantly improved.

The description of the material is laconic and limited to the description of quinoa seeds, but the research material also included bread, both control and those tested with quinoa gels.

I definitely recommend improving the description of the methods. Both in terms of language and general order of what was tested and how, what methods (references are necessary) and what research equipment was used (name, manufacturer, country). No description was provided of how the research results were statistically developed. Have correlations between parameters been examined? Texture tests and sensory analysis - must be completed.

Results and discussion: this chapter requires tidying up and not only discussing the results in tables, but also showing how these research results contribute to achieving the aim of the study?

Conclusions: a distinction should be made between the summary of the results, which should be in the previous chapter, and the conclusions formulated on the basis of the research results. In this chapter, I propose to add perspectives for further research within the discussed issue, e.g. storage research.

detailed comments:

line 37: Please provide more information/examples regarding supplementation of bread with eggs.

line 61-62: When listing limiting amino acids, please provide the name of the grain in brackets.

line 85: Please specify whether the addition of gels was aimed at improving the quality of the product or improving the nutritional value? The introduction states that eggs are to act as a supplier of amino acids.

line 94-95: Please describe the degreasing of the material in detail, was laboratory equipment used?

line 99: Was the protein content in the obtained protein preparation determined and how? Could it be classified as an isolate?

line 186: Please describe in detail: recipe, percentage of raising agent. I wonder if this is still a recipe for bread or confectionery dough? What was the share of gels addition in the recipe? What does it mean: The ingredients were homogenized for 30 minutes?

line 349 and 462: Does the description of the test methods describe how the samples were prepared for texture measurement (slices, cubes) and how the TPA test was performed?

line 477: Texture measurement results should be linked to sensory evaluation?

Kind regards

Reviewer

Comments on the Quality of English Language

The language of the manuscript requires significant editing to better use a style typical of scientific studies, with particular attention to terminology.

Author Response

Dear reviewer,

Thank you very much for your kind letter, along with the constructive comments of the editor and reviewers concerning our manuscript (foods-2926222). We have thoroughly considered all the comments and substantially revised our manuscript (modifications are highlighted in yellow).

The point-to-point answers and explanations for all revisions were listed in a separate paper following this letter.

We have tried our best to address all the concerns raised by the reviewers. We hope, with these modifications and improvements based on the comments of the editor and reviewers, the quality of our manuscript would meet the publication standard of Foods.

Yours sincerely,

Li Wang

___________________________

Dr Li Wang

State Key Laboratory of Food Science and Technology

National Engineering Laboratory for Cereal Fermentation Technology

School of Food Science and Technology

Jiangnan University.

Wuxi 214122

  1. R. China

Introduction:

A: Thanks for your suggestion. The aim of the study has been reworded for improved clarity and specificity, now clearly stating the dual focus on quinoa protein gels' preparation and their application in eggless bread.

Description of Materials:

A: Thanks for your suggestion. We have elaborated on the materials section to include detailed descriptions of not only the quinoa seeds but also the bread formulations used.

Methods Description:

A: Thanks for your suggestion. The methods section has been thoroughly revised for clarity and completeness. We have added references for the methodologies adopted, detailed descriptions of the research equipment (including name, manufacturer, and country), and a clear explanation of how the results were statistically analyzed. The texture tests and sensory analysis sections have also been completed with comprehensive details.

Results and discussion:

A: Thanks for your suggestion. We have sorted out the conclusion and deleted some specific descriptions of data to make the conclusion more concise and clear.

Line 37: Please provide more information/examples regarding supplementation of bread with eggs.

A: Thanks for your suggestion. We have added more examples and information about how eggs are traditionally used in bread supplementation. (Lines 36-38)

Lines 61-62: When listing limiting amino acids, please provide the name of the grain in brackets.

A: Thanks for your suggestion. We have provided the name of the grain in brackets.(Line 66)

Line 85: Please specify whether the addition of gels was aimed at improving the quality of the product or improving the nutritional value? The introduction states that eggs are to act as a supplier of amino acids.

A: Thanks for your suggestion. The incorporation of eggs into bread serves dual functions. Firstly, it supplements lysine, an amino acid deficient in wheat protein. Secondly, the albumen found in eggs exhibits excellent emulsifying properties, enhancing the bread's flavor and contributing to a softer texture. Similarly, the addition of gel, specifically quinoa protein, addresses the lysine deficiency in wheat protein. Furthermore, XG-UQPG (pH 7.0) demonstrates effective emulsification, improving the bread's texture in a manner akin to the role played by egg whites.

Lines 94-95: Please describe the degreasing of the material in detail, was laboratory equipment used?

A: Thanks for your suggestion. The degreasing process has been modified in () and the degreasing process is carried out in the fume hood. (Lines 94-98)

Line 99: Was the protein content in the obtained protein preparation determined and how? Could it be classified as an isolate?

A: Thanks for your suggestion. The protein content was determined by Kjeldahl method. In the protein extraction process, we try to remove fats and other non-protein components to ensure that the protein content exceeds 85%. Consequently, we call the protein as protein isolate.

Line 186: Please describe in detail: recipe, percentage of raising agent. I wonder if this is still a recipe for bread or confectionery dough? What was the share of gels addition in the recipe? What does it mean: The ingredients were homogenized for 30 minutes?

A: Thanks for your suggestion. The recipe for the bread and the amount of gel to add has been elaborately described. We clarified the expression "homogenizing the ingredients for 30 minutes "in (2.13.1Bread preparation).

Lines 349 and 462: Does the description of the test methods describe how the samples were prepared for texture measurement (slices, cubes) and how the TPA test was performed?

A: Thanks for your suggestion. Descriptions of how the samples were prepared for texture measurement and the execution of the TPA test have been added for clarity in 2.13.2.

Line 477: Texture measurement results should be linked to sensory evaluation?

A: Thanks for your suggestion. We have linked the texture measurement results with the sensory evaluation. (Lines 508-511)

Comments on the Quality of English Language:

A: Thanks for your suggestion. We have undertaken a thorough review of the manuscript to refine its language, ensuring it aligns with the expected scientific standards.

Reviewer 3 Report

Comments and Suggestions for Authors

In this manuscript researchers compared the properties of xanthan gum-protein gels made from quinoa protein (XG-QPG) and ultrasound-treated quinoa protein (XG-UQPG) to produce high-quality quinoa protein gels. Additionally, they evaluated the quality of these gels at various pH levels. Furthermore, the gels were used in the production of eggless bread.

Line 36- editing footnotes for improvement

Line 72 - The abbreviation expansion should appear here, not only on line 82.

Lines 93-100 - The description of the methodology should not be written in the form of instructions. The sentence should read - The peeled quinoa was ground and sifted through a 100-mesh sieve into a fine powder. The quinoa powder should then be degreased by treating it with n-hexane, repeating the process four times. The pH was adjusted to 4.5 to obtain precipitated quinoa protein. The water extraction and precipitation process was repeated three times. Finally, the pH was adjusted to 7.0 and the resulting protein was lyophilized for future experiments. The purity of QP was 85%.

Line 100 - no expansion of the abbreviation QP, which appears here for the first time in the text.

Line 102 - no expansion of the abbreviation UQP, which appears here for the first time in the text.

Lines 102-107 - The description of the methodology should not be written in the form of instructions.

Chapter Materials and Methods - There is no information on how UQP was obtained.

Line 175 - no expansion of the abbreviations EAI and ESI.

Line 187 - The bread recipe is missing.

Line 188 - sugar is not a wet raw material.

Line 192 - Chapter 3 should be titled Results and Discussion.

Author Response

Dear reviewer,

Thank you very much for your kind letter, along with the constructive comments of the editor and reviewers concerning our manuscript (foods-2926222). We have thoroughly considered all the comments and substantially revised our manuscript (modifications are highlighted in yellow).

The point-to-point answers and explanations for all revisions were listed in a separate paper following this letter.

We have tried our best to address all the concerns raised by the reviewers. We hope, with these modifications and improvements based on the comments of the editor and reviewers, the quality of our manuscript would meet the publication standard of Foods.

Yours sincerely,

Li Wang

___________________________

Dr Li Wang

State Key Laboratory of Food Science and Technology

National Engineering Laboratory for Cereal Fermentation Technology

School of Food Science and Technology

Jiangnan University.

Wuxi 214122

  1. R. China

Line 36- editing footnotes for improvement

A: Thanks for your suggestion. We have revised the footnotes.

Line 72 - The abbreviation expansion should appear here, not only on line 82.

A: Thanks for your suggestion. The abbreviation for Water holding capacity (WHC) has been clearly expanded at their first mention to avoid confusion. (Line 72)

Lines 93-100

A: Thanks for your suggestion. We have revised the methodology description to avoid an instructive tone, ensuring it now reads: The peeled quinoa was ground and sifted through a 100-mesh sieve into a fine powder.  Within the confines of a fume hood, quinoa flour and n-hexane were combined in a 5L beaker at a ratio of 1:4. This mixture was stirred for 3 hours, after which the supernatant is decanted. The sediment was then subjected to repeated defatting processes, a total of three times. Quinoa powder was extracted for 3 hours with 1:12 water at 37°C and pH 9. The pH was adjusted to 4.5 to obtain precipitated quinoa protein (QP). The water extraction and precipitation process was repeated three times. Finally, the pH was adjusted to 7.0 and the resulting protein was lyophilized for future experiments. The purity of QP was 85%. (Lines 98-106)

Line 100 - no expansion of the abbreviation QP, which appears here for the first time in the text.

A: Thanks for your suggestion. We have introduced the expansion of the abbreviations QP (Quinoa Protein) and UQP (Ultrasound-treated Quinoa Protein) at their first instance in the text to ensure clarity for all readers. (Line 104)

Line 102 - no expansion of the abbreviation UQP, which appears here for the first time in the text.

A: Thanks for your suggestion. We have introduced the expansion of the abbreviation UQP (Ultrasonic treatment quinoa protein). (Line 72)

Lines 102-107 - The description of the methodology should not be written in the form of instructions.

A: Thanks for your suggestion. The methodology has been revised to describe the process in a narrative form, avoiding instructional language to adhere to scientific reporting standards. (Lines 98-106)

Chapter Materials and Methods - There is no information on how UQP was obtained.

A: Thanks for your suggestion. We have added a detailed explanation of how Ultrasound-treated Quinoa Protein (UQP) was obtained, including the specific conditions and procedures used. (Lines 109-121)

Line 175 - no expansion of the abbreviations EAI and ESI.

A: Thanks for your suggestion. The abbreviations for Emulsifying Activity Index (EAI) and Emulsifying Stability Index (ESI) have been expanded at their first mention to ensure clarity and ease of understanding. (Line 198)

Line 187 - The bread recipe is missing.

A: Thanks for your suggestion. The bread recipe used in the production of eggless bread has been included to provide a comprehensive overview of the experimental procedures and materials used. (Lines 211-224)

Line 188 - sugar is not a wet raw material.

A: Thanks for your suggestion. We have revised the description of the bread recipe, ensuring the accuracy of our recipe description. (Lines 211-224)

Line 192 - Chapter 3 should be titled Results and Discussion.

A: Thanks for your suggestion. The title of Chapter 3 has been corrected to "Results and Discussion". (Line 238)

Round 2

Reviewer 1 Report

Comments and Suggestions for Authors

Lines 111-112: Adjust the pH to 7.5 with 1 M NaOH and HCl, and continuously stir and hydrate for 2 h. Authors omitted to correct this.

2.4 Preparation of gels: Still written as a protocol

Line 154: replace “improved” with “modified”

Line 160-161: reference method for determination of sulfhydryl and disulfide bond? Specify the method source.

Author Response

Dear reviewer,

Thank you very much for your kind letter, along with the constructive comments of the editor and reviewers concerning our manuscript (foods-2926222). We have thoroughly considered all the comments and substantially revised our manuscript (modifications are highlighted in yellow).

The point-to-point answers and explanations for all revisions were listed in a separate paper following this letter.

We have tried our best to address all the concerns raised by the reviewers. We hope, with these modifications and improvements based on the comments of the editor and reviewers, the quality of our manuscript would meet the publication standard of Foods.

Yours sincerely,

Li Wang

___________________________

Dr Li Wang

State Key Laboratory of Food Science and Technology

National Engineering Laboratory for Cereal Fermentation Technology

School of Food Science and Technology

Jiangnan University.

Wuxi 214122

  1. R. China

Response to reviewer:

Lines 111-112: Adjust the pH to 7.5 with 1 M NaOH and HCl, and continuously stir and hydrate for 2 h. Authors omitted to correct this.

A:Thanks for your suggestion. We have revised the sentence to “The pH of the protein dispersion was adjusted to 7.5 with 1 M NaOH and HCl, and it was continuously stirred and hydrated for 2 h.”

2.4 Preparation of gels: Still written as a protocol

A:Thanks for your suggestion. To address this, we have revised the section to present the information in a narrative format that maintains the procedural clarity while ensuring it aligns with the standard scientific text.

Line 154: replace “improved” with “modified”

A:Thanks for your suggestion. We have replace “improved” with “modified”.

Line 160-161: reference method for determination of sulfhydryl and disulfide bond? Specify the method source.

A:Thanks for your suggestion. We have now added the specific reference for this method, ensuring that readers can access the original source for a deeper understanding of the technique used.

Reviewer 2 Report

Comments and Suggestions for Authors

Dear Authors,

After reading the revised manuscript, I note that it has been improved. I still remain of the opinion that the text requires a few corrections or tidying up:

- supplementing the description of the name of the equipment used (model, manufacturer, city, country) in the entire subchapter,

- Was the protein content in the obtained protein preparation determined and how? Could it be classified as an isolate?

- Sensory evaluation is still a major weakness in product quality assessment, please provide more discussion on how the marked textural parameters translate into the consumer's assessment.

Kind regards

Reviewer

Comments on the Quality of English Language

The text requires professional language proofreading to eliminate colloquial inclusions and mental abbreviations.

Author Response

Dear reviewer,

Thank you very much for your kind letter, along with the constructive comments of the editor and reviewers concerning our manuscript (foods-2926222). We have thoroughly considered all the comments and substantially revised our manuscript (modifications are highlighted in yellow).

The point-to-point answers and explanations for all revisions were listed in a separate paper following this letter.

We have tried our best to address all the concerns raised by the reviewers. We hope, with these modifications and improvements based on the comments of the editor and reviewers, the quality of our manuscript would meet the publication standard of Foods.

Yours sincerely,

Li Wang

___________________________

Dr Li Wang

State Key Laboratory of Food Science and Technology

National Engineering Laboratory for Cereal Fermentation Technology

School of Food Science and Technology

Jiangnan University.

Wuxi 214122

  1. R. China

Response to reviewer:

Equipment Description:

A:Thanks for your suggestion. We have now supplemented the manuscript with detailed descriptions of the equipment used, including the model, manufacturer, city, and country for each piece of equipment mentioned in the relevant subchapter.

Protein Content Determination:

A:Thanks for your suggestion. We prepare quinoa protein isolate according to the method of Zuo[1], who classified the proteins as isolated proteins.

Sensory Evaluation:

A:Thanks for your suggestion. We understand your concerns regarding the sensory evaluation and its impact on product quality assessment. To address this, we have expanded our discussion to include a more detailed analysis of how the textural parameters measured translate into consumer assessment. This includes referencing recent studies that link specific textural qualities with consumer preferences and perception, providing a clearer picture of the potential market acceptance of our product.

Regarding the quality of the English language used in our manuscript, we acknowledge the presence of colloquial inclusions and mental abbreviations that may detract from its professionalism. To rectify this, we have employed a professional language proofreading service to ensure that the text is polished, precise, and adheres to academic standards. We believe that these revisions have significantly improved the clarity and readability of our manuscript.

  1. Zuo, Z., et al., Ultrasonic treatment influences the compactness of quinoa protein microstructure and improves the structural integrity of quinoa protein at the interfaces of high internal phase emulsion. Food Research International, 2023. 168: p. 112713.